# Innovative Antibiofilm Smart Surface against *Legionella* for Water Systems

**DOI:** 10.3390/microorganisms10050870

**Published:** 2022-04-21

**Authors:** Simona Filice, Emanuele Luigi Sciuto, Silvia Scalese, Giuseppina Faro, Sebania Libertino, Domenico Corso, Rosario Manuel Timpanaro, Pasqualina Laganà, Maria Anna Coniglio

**Affiliations:** 1Istituto per la Microelettronica e Microsistemi-Consiglio Nazionale delle Ricerche (CNR-IMM), Ottava Strada 5, 95121 Catania, Italy; simona.filice@imm.cnr.it (S.F.); silvia.scalese@imm.cnr.it (S.S.); sebania.libertino@imm.cnr.it (S.L.); domenico.corso@imm.cnr.it (D.C.); ma.coniglio@unict.it (M.A.C.); 2Azienda Ospedaliero Universitaria Policlinico “G. Rodolico-San Marco”, Via S. Sofia 78, 95123 Catania, Italy; manueltimpanaro@gmail.com; 3Azienda Sanitaria Provinciale di Catania, Via S. Maria La Grande 5, 95124 Catania, Italy; giuseppina.faro@aspct.it; 4Regional Reference Laboratory of Clinical and Environmental Surveillance of Legionellosis, Department of Biomedical and Dental Sciences and Morphofunctional Imaging, Torre Biologica 3p, AOU ‘G. Martino’, University of Messina, Via C. Valeria, S.N.C., 98125 Messina, Italy; plagana@unime.it; 5Regional Reference Laboratory of Clinical and Environmental Surveillance of Legionellosis, Department of Medical and Surgical Sciences and Advanced Technologies “G.F. Ingrassia”, University of Catania, Via Sofia 87, 95123 Catania, Italy

**Keywords:** *Legionella pneumophila*, water safety plan, biofilm, smart coatings, s-PBC

## Abstract

*Legionella pneumophila* contamination of water systems is a crucial issue for public health. The pathogen is able to persist in water as free-living planktonic bacteria or to grow within biofilms that adhere to and clog filters and pipes in a water system, reducing its lifespan and, in the case of hospital buildings, increasing the risk of nosocomial infections. The implementation of water management is considered to be the main prevention measure and can be achieved from the optimization of water system architecture, notably introducing new materials and strategies to contrast *Legionella* biofilm proliferation and so prolong the water system functionality. In this research, we propose a new smart surface against *L. pneumophila* biofilm formation. This is based on an innovative type of coating consisting of a sulfonated pentablock copolymer (s-PBC, commercially named Nexar™) deposited on top of a polypropylene (PP) coupon in a sandwich filter model. The covering of PP with s-PBC results in a more hydrophilic, acid, and negatively charged surface that induces microbial physiological inhibition thereby preventing adhesion and/or proliferation attempts of *L. pneumophila* prior to the biofilm formation. The antibiofilm property has been investigated by a Zone of Inhibition test and an in vitro biofilm formation analysis. Filtration tests have been performed as representative of possible applications for s-PBC coating. Results are reported and discussed.

## 1. Introduction

The *Legionella pneumophila* colonization of water systems represents one of the main issues for public health, especially referring to those contaminations that occur in buildings, such as hospitals and other healthcare environments, hosting fragile persons who are particularly susceptible to infections and associated clinical complications [1]. Water safety plans (WSPs) are designed to reduce the health risks associated with water systems. Understanding where the risks exist throughout the water system and how the quality of the water can be maintained is crucial to creating a good WSP. To this purpose, it is imperative to identify where the potential risks are coming from, as well as to implement measures to reduce the identified risks as much as possible [2].

*Legionella* is a pathogen widely diffused in nature whose eradication for clinical prevention is still challenging [3]. When colonization occurs, in fact, *Legionella* finds the optimal environmental conditions to survive, including the water temperature and stagnation, availability of organic sediments and presence of free-living protozoa, that protect bacteria from disinfection procedures. This survival of *Legionella* can be accomplished by biofilm formation. Biofilm is a complex aggregation of microorganisms encapsulated inside a matrix of secreted extracellular polymeric substances (EPSs) [4], resulting from an interaction of planktonic bacteria with a generic surface, such as in premise plumbing or that of filters used for water depuration. The electrostatic forces operating between the bacterial membrane and the target surface induce their attachment and subsequent EPSs secretion that increases the interaction strength. Once aggregated, the microorganisms start colonizing the surface and growing inside the EPSs matrix, completing the biofilm architecture. At a high mass level, then, the bacteria can be released and dispersed outside the biofilm in a detachment process that increases the level of pathogen diffusion in water and the risk of infection [5]. 

It has been demonstrated that *Legionella* inside a biofilm matrix can be resistant to biocide treatments, affecting the efficiency of most common water disinfection measures [6,7]. The strategies adopted so far to limit biofilm formation in water systems, in fact, have been mostly focused on the modification of water chemical composition [8,9]. This can result from the addition of biocides, such as oxidizing disinfectants, surfactants, and antibiotics, that affect the bacteria physiology before and after the biofilm formation [10]. However, the biofilm itself can block the biocide molecules from diffusing inside and hence being effective against the encapsulated microorganisms. In addition, disinfectants may not reach distal areas of water systems and can dissipate throughout the plumbing, making them less effective so that a secondary chemical water disinfection process could be needed. This could lead to the use of a large amount of chemicals to achieve a complete disinfection, resulting in high costs and in the dispersion of unsafe compounds, such as disinfection by products (DBP) [11].

Therefore, in recent years new strategies have been developed to be used in synergy with primary chemical treatments in order to enhance the efficiency of common water disinfection methods and reduce the consumption of chemicals. Among these, the introduction of smart surfaces is an important step forward [12] since, being applied as coatings for water pipes and filters, they can increase the lifespan of a water system. The aim of coatings is, in fact, to block the bacterial attachment and/or proliferation, needed for biofilm formation, on surfaces of plumbing fittings and to prevent the clogging of filter meshes and the obstruction of pipes.

In recent decades a lot of smart surfaces have been made and employed as coatings for water filters [13,14] providing a physical barrier between *Legionella* or other water-borne bacteria, and fragile persons. These filters, in particular, can be regarded as additional barriers in case of low water flow areas or dead legs, that cannot be reached by chemicals. Thus, they are needed for each point of use in a water system but if not regularly replaced they can be colonized by bacteria such as *Legionella* [15]. For this reason, the use of coated filters with a prolonged lifespan and an antibiofilm effect, which require fewer change-outs, could provide a cost-effective method to prevent Legionnaire’s disease [16] and to increase the efficiency of water cleaning. Furthermore, the need for methods to reduce *Legionella* colonization of piping requires the development of single or multi-level surface functionalization steps, preferably before the biofilm is formed [17].

Smart coatings can hinder biofilm formation in at least two different ways: (i) some of them can prevent bacteria from adhering, creating a steric, mechanic, and/or electrostatic barrier [18,19,20]; (ii) others have a bactericidal effect towards microorganisms before or after contact with the surface coating [21,22].

Many types of filters coatings have been reported to date. Among them, electrically heatable carbon nanotube (CNT) functionalization of point-of-use (POU) filters demonstrated a removal rate of 99.9% of *L. pneumophila* in water [23]. Graphene oxide quantum dots (GOQDs) were used to create an antibacterial surface with reasonably strong antibiofouling properties [24]. The sulfonated pentablock copolymer (s-PBC), commercially named Nexar™, has also been reported for water disinfection and filtration [25]. In particular, the s-PBC has shown a controlled swelling and good mechanical properties in the hydrated state together with a good level of functionalization, processability, and low cost. It was demonstrated that the polymer was able to induce the death of *P. aeruginosa* by a contact killing mechanism due to water acidification induced by the polymer, preventing microbial adhesion and replication. Furthermore, other works reported the use of s-PBC in water purification applications as adsorbent material for heavy metals removal [26] or in combination with known photocatalysts for azo dye degradation [27,28].

Considering the above reported evidence, we proposed the Nexar^TM^ polymer as smart coating for antibiofilm filter in water systems. The aim was to prevent the biofilm formation on a water filter surface avoiding its progressive deterioration due to microbial proliferation and meshes clogging. We investigated the polymer after deposition on a polypropylene (PP) substrate, a material that is commonly used for water filters. The antibiofilm activity of the modified surface was tested through a biofilm induction experiment and a physiological inhibition susceptibility (Zone of Inhibition) test, both performed on *Legionella pneumophila* serogroup (SG) 2–16. 

*L. pneumophila* SG 1 to 10, 12, 13, and 15 have always been isolated from water distribution systems worldwide more frequently than the other serogroups [29,30,31]. Although environmental strains (SG 2–15) account for only 16 to 20% of legionellosis cases [32], there is evidence that patients with *L. pneumophila* of SG 2 to 15 show typical symptoms of *Legionella* pneumonia, but *Legionella* urinary antigen tests (UATs), which are rapid tools for early diagnosis of legionellosis, are negative because UATs detect only SG 1 [33,34,35]. From a public health perspective, this is noteworthy because there is an association between delayed laboratory diagnosis, therapy, and prognosis. In particular, the delay in appropriate therapy for legionellosis is associated with increased mortality [36]. For these reasons, we decided to test only environmental strains (SG 2–16).

Another aim was to consider a possible application of the s-PBC as smart coating for water filters to prevent the persistence of bacteria in water. The obtained results confirmed the microbial inhibition properties of Nexar™ and its suitability as smart coating for water cleaning.

## 2. Materials and Methods

### 2.1. Chemicals

*Legionella pneumophila* SG 2–16 strain has been isolated from drinking water. Tryptic Soy Broth (TSB) was purchased from Sigma (St. Louis, MO, USA). Glycine Vancomycin Polymyxin Cycloheximide (GVPC) agar plates and LIVE/DEAD Baclight Bacterial Viability kit were purchased from Thermo-Fisher (Waltham, MA, USA). Plastic Petri dishes (Ø 18 cm) were from Aptaca s.r.l. Glass flasks of 200 mL were from Simax. Multilayer polypropylene (PP) filters were produced in-house using the melt-blown technology process as reported by Sikorskaet al. [37]. The sulfonated pentablock copolymer poly(tBS–HI–sS:S–HI–tBS) solution, or s-PBC, with 10–12 wt% polymer in a cyclohexane/heptane mixed solvent was provided by courtesy of Kraton Polymers LLC (Houston, TX, USA). The 4 wt% solution of s-PBC for filters depositions was prepared by dispersing the commercial s-PBC solution in a polar solvent (isopropyl alcohol, IPA).

### 2.2. Zone of Inhibition Test: Coupons and L. pneumophila Culture Preparation

Polypropylene (PP) coupons (1 cm diameter) were sterilized in an autoclave for 15 min at 121 °C. Then, 0.2 mL of s-PBC polymer solution was deposited on top of sterile PP coupons and left drying for 24 h at room temperature. Then, both deposited (s-PBC@PP) coupon and PP coupon were washed by immersion in 70% ethanol for 10 min and rinsed in sterile distilled water (dH_2_O) and left drying overnight at room temperature. In parallel, *L. pneumophila* serogroup (SG) 2–16 was grown overnight in 10 mL of Tryptose broth at 37 °C. The resulting culture was, then, diluted in 5 mL of sterile dH_2_O and 0.1 mL of this dilution was spread over a selective GVPC agar plate and left drying for 30 min at room temperature. Two drops of 0.2 mL of sterile dH_2_O were spotted on a plate and both PP and s-PBC@PP coupons were applied on top of the drops, with the active face down. The importance of filling the space between coupons and agar medium with water has been previously demonstrated [25]. The plate was incubated at 37 °C and colonies were analysed after 24 h.

### 2.3. Biofilm Formation Test: Sample Preparation

*L. pneumophila* SG 2–16 culture for biofilm analysis experiment was prepared as follows: colonies of *L. pneumophila* were grown on GPVC plate overnight at 37 °C and, then, suspended in Tryptose broth until the OD490was about 0.6. The resulting bacterial suspension was then diluted 1:6 in fresh Tryptose broth and incubated again at 37 °C with 5% CO_2_ for approximately 3 h, in order to reach the mid-log phase. The mid-log was, then, diluted 1:2500 in fresh pre-warmed fresh broth and 0.7 mL of dilution were spotted into each well of a 4-well chamber slide, reported in Figure 1a. In parallel, a 0.5 × 1 cm PP coupon was put on top of *L. pneumophila* culture in chamber 2 while two 0.5 × 1 cm s-PBC@PP coupons were deposited in chambers 3 and 4, as shown in Figure 1b. Both PP and s-PBC@PP coupons had been prepared as described before. Once prepared with culture and coupons, the chamber slide was incubated at 37 °C with 5% CO_2_. After approximately 16 h, the culture was aspirated from the corner of each chamber and 0.7 mL of fresh pre-warmed broth was dispensed all along the wall of the chambers, in order to avoid shear forces that could disrupt the biofilm. This step was repeated after 24 h.

### 2.4. Biofilm Formation Test: Sample Analysis

After 4 days of incubation at 37 °C with 5% CO_2_, the biofilm formation inside the chamber slide was checked as follows: *L. pneumophila* 2–16 culture was discarded from all wells and these were cleaned twice using 0.7 mL of sterile saline solution. Then, 0.7 mL of 10% formalin was added at room temperature for 30 min, in the dark, to fix the *L. pneumophila* cells inside the chambers. Once washed again with saline solution, chambers were treated with LIVE/DEAD Baclight mix of fluorescent dyes (from Thermo-Fisher), added at room temperature for 15 min in the dark, to selectively stain bacteria including those eventually attached to the floating surface of loaded coupons. Once the staining solution discarded and washed again, both lid and chamber components of the chamber slide were removed keeping the glass slide on bottom and the PP and s-PBC@PP coupons that were analysed by fluorescence microscopy using a 488 nm light source on top.

### 2.5. PH Measurements

The pH measurements were performed using a GLP-22 pH meter (GIBSON) using the following samples: sterile tap and distilled water exposed to floating PP and s-PBC@PP coupon; fresh buffered broth without bacteria exposed to floating s-PBC@PP coupon; *L. pneumophila* SG 2–16 culture exposed to s-PBC@PP coupon in 1.5 mL centrifuge tube, used as control; *L. pneumophila* SG 2–16 culture exposed to PP and s-PBC@PP coupons inside the chamber slide, used in the biofilm formation experiment. All samples were analysed before and after 10 min and 1; 16; 48 h exposure to floating coupons in 0.7 mL volume.

Samples collected from the chamber slide were measured only after 16 h of incubation, i.e., when the *L. pneumophila* culture was substituted with fresh broth, in order to keep the system sterile during the incubation at 37 °C–5% CO_2_ in the biofilm formation experiment.

### 2.6. Filter Coating Test: Sample Preparation

Then, 1L of *L. pneumophila* 2–16 water culture was prepared by adding a known concentration of bacteria in sterile tap water. The filtration was performed using a 4.7 cm diameter s-PBC@PP and PP filter, used as reference. The PP filters are composed of fibres with diameter ranging between 0.2 and 5 µm. They form a three-dimensional network where the pore size has a wide range distribution. It is thus more appropriate to define a porosity instead of a pore size. The filter porosity ε_F_ is 0.98 and was determined using the formula: ε_F_ = 1 − ρ_SF_/(ρ_F_·L), where ρ_SF_ is the surface density calculated as the mass of the filter divided by the surface area (m_F_/A_F_); ρ_F_ is the fibres’ material density (910 Kg/m^3^ for polypropylene was used); L is the PP filter thickness.

Filters were inserted into a plastic reusable filtration unit of 250 mL (from Nalgene™) conventionally used for water analysis, shown in Figure 2. For each type of filter, i.e., PP and s-PBC@PP, five filtrations of 50 mL and one last filtration of 200 mL of *L. pneumophila* culture were performed. The water flux measured was 18.77 mL/min for PP filter and 2.93 mL/min for the one coated by s-PBC. This strong reduction is due to the compact nature of the covering layer whose pores are smaller than the mesh of the PP filter. The culture was analysed before and after all filtrations by plating 0.1 mL of both samples into selective GVPC agar and allowed to grow at 37 °C–5% CO_2_ for 7 days. Bacteria remaining on filter were analysed by immersing the filter into 40 mL of sterile distilled water and, then, vortexing for 2 min so that all bacteria were released and suspended. A volume of 0.1 mL of this sample was then plated on selective agar medium and incubated at 37 °C–5% CO_2_ for 7 days.

## 3. Results

The antibiofilm effect of s-PBC coating was tested on *L. pneumophila* SG 2-16. A first analysis was performed on bacteria plated in a GVPC agar substrate. The Inhibition Zone test, reported in Figure 3, showed that after 24 h a 2-cm diameter microbial inhibition zone appeared all around the s-PBC@PP coupon (Figure 3a). As explained in our previous work [25], this could be due to the acidification effect of s-PBC polymer that could block bacteria proliferation on selective medium. The absence of such a zone around the PP coupon (Figure 3b), by contrast, confirmed a regular proliferation of bacteria.

In addition, the smart coating was investigated on broth culture of *L. pneumophila* in order to study its ability to prevent biofilm formation as evidenced by the coupon meshes staying clean.

The analysis was performed on a chamber slide system (see Section 2) and revealed that after 4 days of incubation at 37 °C with 5% CO_2,_
*L. pneumophila* formed a biofilm on the bottom of empty chambers and in floating PP coupons as shown in Figure 4a–f. The biofilm produced a green fluorescence signal since it is composed of live cells that absorbed the SYTO 9 dye of LIVE/DEAD Baclight kit, as proved by the detail at 100× magnification reported in Figure 4c*.

By contrast, no biofilm was formed in floating s-PBC@PP coupons (Figure 4g–i). In this case, the coupon fibres, that were orange-stained due to the propidium iodide labelling of altered cells, appeared free of any coverage as shown at 40× and 100× magnification.

As previously reported in the Inhibition Zone test results, we supposed that the bacteria inhibition was induced by acidification of water due to the s-PBC coating. For this reason, in order to deeper investigate the inhibition mechanism, the pH variation was tested in all types of media, and samples used in both the zone of inhibition (H_2_O and dH_2_O) and biofilm formation (all media) test (Table 1). No changes were observed in samples exposed to PP coupon, as expected. By contrast, the pH decreased heavily in distilled water (7.28 to 2.95) and less in tap water (8.7 to 6.7) exposed to the s-PBC@PP coupon, while it was quite stable in all other samples containing Tryptose broth. This was consistent with the fact that the broth used to prepare *L. pneumophila* cultures was buffered at pH 7. Although the pH of the whole solution remained close to neutrality, the *Legionella* inhibition was observed in broth and tap water after exposure to s-PBPC@PP in the biofilm formation test (see before). This led to the hypothesis that the antibiofilm effect could be influenced by the bacteria interactions with the acid surface of the s-PBC@PP coupon, the releases of H^+^ species being responsible for cell inhibition, and not the direct acidification of water volume as observed for distilled water.

Having proved its inhibition effect on bacteria physiology, the smart s-PBC polymer was tested as a potential coating for water filters. Results from serial filtrations of *L. pneumophila* contaminated tap water samples are reported in Figure 5. 

Figure 5 reports the colony density observed on *L. pneumophila* after filtrations. Five aliquots of bacteria solutions were filtered using a PP filter or an s-PBC@PP one, and 0.1 mL of each filtrate was used for the culture.

Relative to the starting density (Figure 5(aI,bI)), the number of colonies decreased passing from the first to the fifth filtration with both PP and s-PBC@PP filters (Figure 5(aII,aIII,bII,bIII), respectively) as an effect of bacteria deposition on their surface. However, the average colony density observed on plates prepared from PP-filtered 50 mL samples was higher than that from s-PBC@PP-filtered samples both at the first filtration test and after five consecutive cycles. This confirms the higher bacteria removal efficiency of s-PBC@PP filter with respect to the commercial one. This effect is probably due to a mesh reduction determined by the polymeric coating that hinders the bacteria from passing through the filter. The same result was observed for the filtration tests conducted with the same filters using another 200 mL aliquot from the starting water sample (Figure 5(cI,cII)), where no colonies appeared in the s-PBC@PP-filtered sample compared to the PP-filtered one. Figure 5d, instead, shows the residual colony density of *L. pneumophila* observed on filter surfaces, and reports a higher density in the s-PBC@PP surface residual (Figure 5(dII)) compared to the PP one, suggesting that the smart surface was able to block the passage of *L. pneumophila* more efficiently.

## 4. Conclusions

In this study, we propose a Nexar^TM^-modified surface for smart *L. pneumophila* removal and simultaneous prevention of its biofilm formation in water systems.

Prevention of *L. pneumophila* contamination is a key component of the water management programs implemented by the water safety plans (WSPs) [38]. Conventional water control techniques are often ineffective against biofilm formation, which is a critical issue for bacterial colonization of plumbing [10]. In particular, it has been estimated that 95% of the biomass in man-made water systems is attached to the walls as biofilm [39,40], and there is evidence that the biofilm’s physical stability is critical for the efficacy of disinfection procedures [41]. Therefore, techniques for the control of biofilm growth can provide an important contribution in overcoming current water treatment limitations, and biofilm management can be regarded as part of an integrated approach to mitigate Legionnaires’ disease incidence. This evidence warrants the need for novel preventive approaches focused on biofilm removal and inactivation systems.

Our data showed that the Nexar^TM^-modified coupons of polypropylene (s-PBC@PP) effectively were able to block *L. pneumophila* SG 2–16 growth, as demonstrated by the inhibition zone which appeared on a GVPC agar 24 h culture. A possible explanation of this inhibition effect is that the acid and negative surface of s-PBC is able to prevent both bacteria adhesion to the filter and its proliferation by acidification of water strictly in contact with the filter surface.

Assays on a chamber slide system showed that the s-PBC@PP coupons were also effective against biofilm formation in large water volumes. This antibiofilm formation occurred while maintaining the pH of the whole bacteria solution close to neutrality (a condition that allows bacteria survival). Thus, the antibiofilm activity is ascribed to a surface effect induced by direct interaction of bacteria with the surface of s-PBC. 

As expected, considering a possible application of the s-PBC as smart coating for water filters, data reported in Figure 5 showed that the polymer was able to block Legionella on the filter surface, preventing the bacteria from persisting in water. In this research, the authors’ aim was not to demonstrate a killing effect of s-PBC towards *L. pneumophila* but to verify its inhibition properties against the bacterial biofilm formation. Therefore, colony counts of the CFU/L in both starting and retention solutions were not performed or matched.

A limitation of the current study is due to the fact that the tested *L. pneumophila* biofilm was performed under laboratory conditions. Thus, a future perspective of the work could include natural more complex biofilms. Moreover, taking into consideration that Legionella species may be able to quickly recolonize the water systems [42], further studies are needed to verify if this antibiofilm effect is totally efficient also after a long period of time.

The aim of the work was to test the antibiofilm activity of the s-PBC. This was verified by testing the anti-attachment process of planktonic Legionella on top of the modified filter surface, as above reported in Figure 4g–i, and by the observation that, although they were randomly deposited by gravity, bacteria were inhibited in their proliferation process, as described in Figure 5(cII). Moreover, another possible application of the proposed polymer could be the coating of pipes as antibiofilm surface.

In conclusion, the proposed technique has been demonstrated to have a great potential in reducing planktonic and sessile (biofilm) Legionella cells, representing a novel preventive approach as an inactivation system and for avoiding biofilm formation.

## Figures and Tables

**Figure 1 microorganisms-10-00870-f001:**
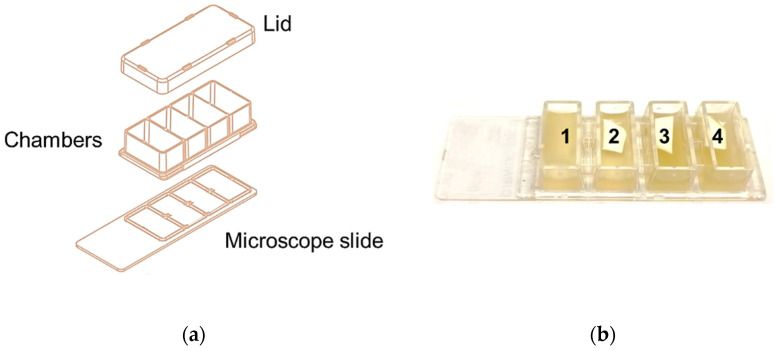
Chamber slide for biofilm analysis. (**a**) Structural components of cultural system. (**b**) Sample loaded: (1) *L. pneumophila* culture; (2) *L. pneumophila* with Floating PP coupon; (3–4) Replicas of *L. pneumophila* culture with floating s-PBC coupon.

**Figure 2 microorganisms-10-00870-f002:**
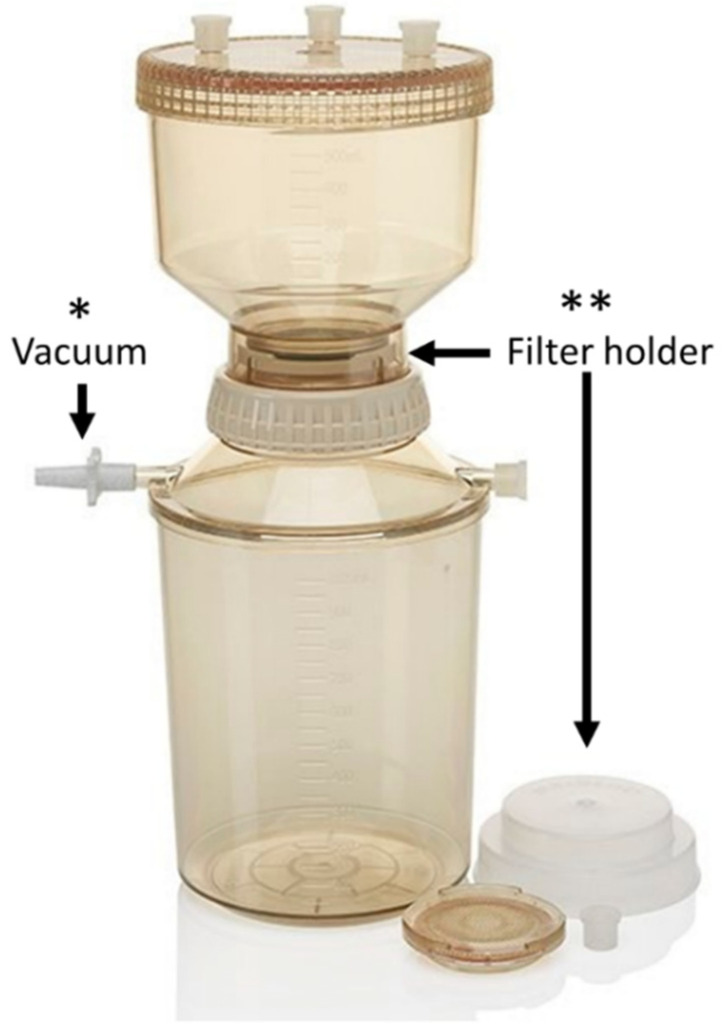
Nalgene^TM^ reusable unit for 250 mL water filtration. Details of vacuum nozzle (*) and filter holder (**).

**Figure 3 microorganisms-10-00870-f003:**
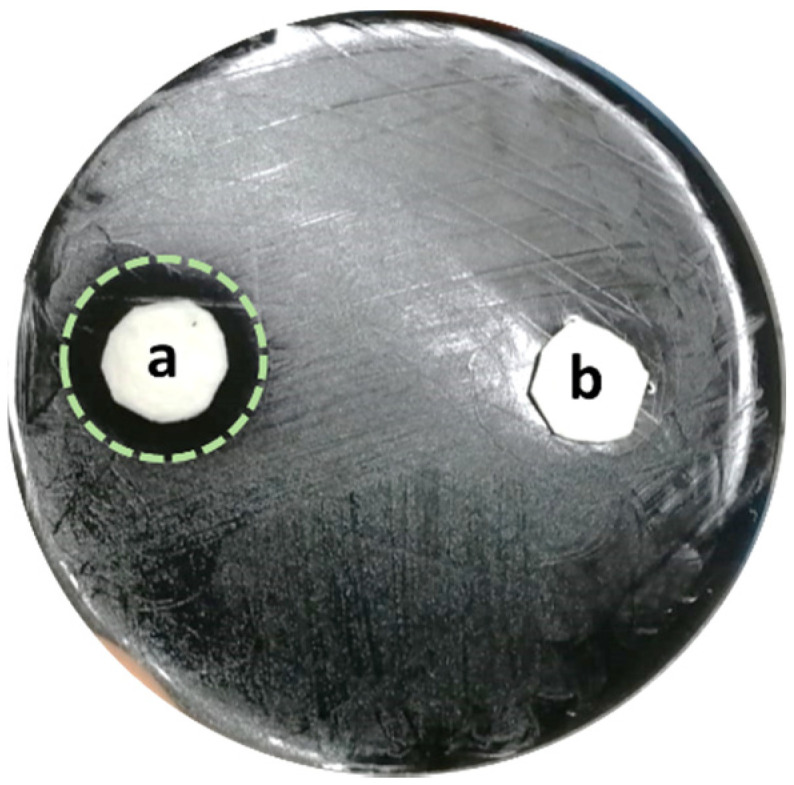
Zone of Inhibition Test on *Legionella pneumophila* SG 2–16: **a** s-PBC@PP coupon surrounded by an inhibition zone (green dashed line); **b** PP coupon.

**Figure 4 microorganisms-10-00870-f004:**
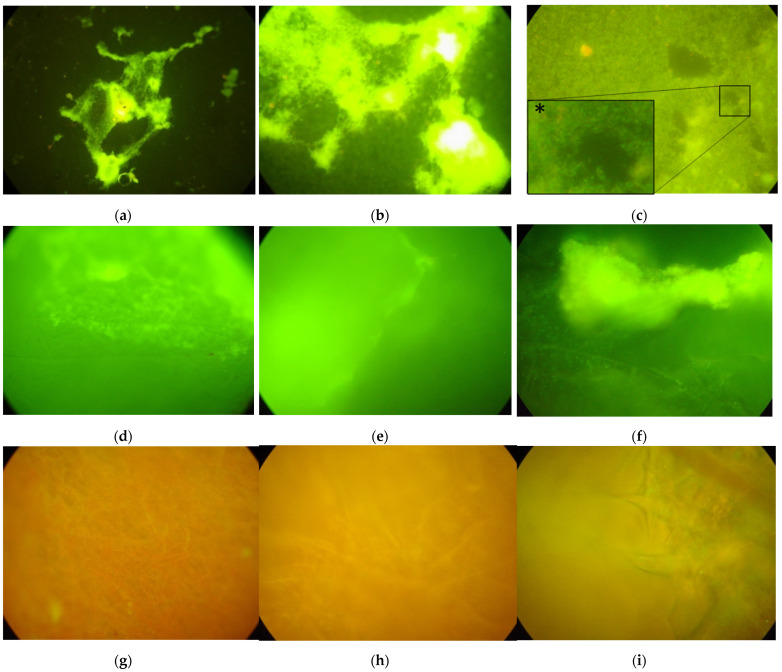
Biofilm of *Legionella pneumophila* SG 2–16 at 10×, 40× and 100× of microscope magnification formed without coupon (**a**–**c**), on floating PP coupons (**d**–**f**) and on floating s-PBC@PP coupons (**g**–**i**), respectively. Detail of cell bodies at 100× magnification (*****).

**Figure 5 microorganisms-10-00870-f005:**
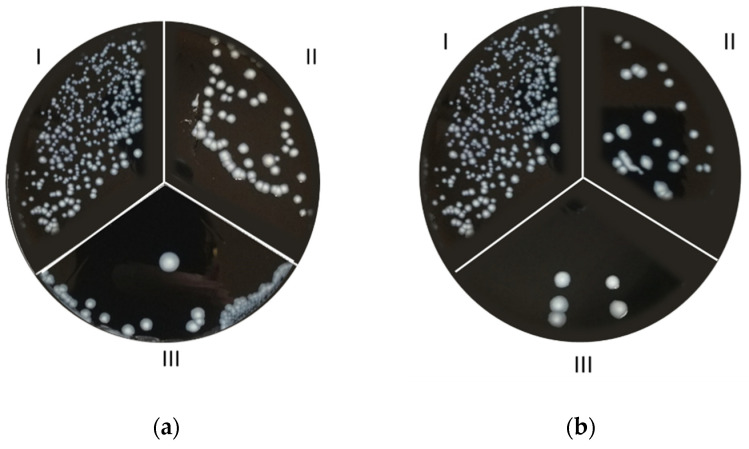
Colony density of *L. pneumophila* in starting (**aI**,**bI**) and filtered water samples: 50 mL sample after 1 (**aII**) and 5 (**aIII**) filtrations with PP filter; 50 mL sample after 1 (**bII**) and 5 (**bIII**) filtrations with s-PBC@PP filter; 200 mL sample after 6 filtrations with PP (**cI**) and s-PBC@PP (**cII**) filter. Colony density of total *L. pneumophila* blocked on PP (**dI**) and s-PBC@PP (**dII**) filter surface.

**Table 1 microorganisms-10-00870-t001:** pH analysis of various media involved in the biofilm formation experiment after exposure to floating coupons.

Sample	T0	10 min	1 h	16 h	48 h
H_2_O + PP	8.73	8.80	8.81	7.48	7.22
H_2_O + s-PBC@PP	8.71	6.83	6.73	6.81	6.70
dH_2_O + PP	7	6.76	6.58	6.25	5.85
dH_2_O + s-PBC@PP	7.28	3.53	3.12	2.98	2.95
Tryptose broth + s-PBC@PP	6.70	6.50	6.45	6.73	6.75
*L. pneumophila* + s-PBC@PP	6.71	6.42	6.37	6.00	6.66
*L. pneumophila* + PP *	6.72	N.D.	N.D.	5.86	N.D.
*L. pneumophila* + s-PBC@PP *	6.71	N.D.	N.D.	6.78	N.D.

* Sample collected from chamber slides. N.D. = Not determined.

## Data Availability

Not applicable.

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
