# Peer review of "Innovative Antibiofilm Smart Surface against Legionella for Water Systems"

_microorganisms, 2022, doi:10.3390/microorganisms10050870_

Round 1

Reviewer 1 Report

My comments regarding the work titled: Innovative antibiofilm smart surface against Legionella for water systems:
1. The effectiveness of the s-PBC system has not been sufficiently proven. 2. The authors used the qualitative zone of inhibition test, which does not prove that the proliferation of the entire bacterial population was inhibited. 3. The study should be performed using quantitative methods. 4. Experiments should be performed in several repetitions, and the results should be statistically analyzed.

Author Response

My comments regarding the work titled: Innovative antibiofilm smart surface against Legionella for water systems:

  1. The effectiveness of the s-PBC system has not been sufficiently proven.

Authors thank the referee for the comments and agree with the referee that the effectiveness of our system has not been sufficiently proven. However, as reported in lines 368-369, our aim was just to prove that the proposed polymer had a ‘potential’ in reducing planktonic and sessile Legionella cells in order to avoid its biofilm formation.  Thus, authors’ aim was not to prove the ‘effectiveness’ of the proposed system – in real conditions – but to verify the its ‘efficacy’ under ideal/experimental circumstances.

  1. The authors used the qualitative zone of inhibition test, which does not prove that the proliferation of the entire bacterial population was inhibited.

Authors apologize if the reported data were not completely clear. The Zone of Inhibition test, that has been used to prove the inhibition properties of the polymer, has been chosen because the bacterial population tested was composed by environmental strains which are supposed to have all the same physiological features, thus, giving the same response.

  1. The study should be performed using quantitative methods.

Authors agree with referee that more tests could be performed. However, the aim of the work was to specifically verify the antibiofilm properties of the proposed polymer, so that it wasn’t necessary to use quantitative methods. Obviously, in further studies, it would be interesting to deeper investigate the antibiofilm activity of the polymer towards more bacterial species, including quantitative methods such as MIC/MBC.

  1. Experiments should be performed in several repetitions, and the results should be statistically analyzed.

The proposed study was just a preliminary assay of the potential applications of s-PBC polymer as antibiofilm coating against Legionella. However, authors had already tested this material (including several replicas) in a previous work (see ref [25]), where the response of bacteria after polymer exposure by Zone of Inhibition test (changing volumes and pH of solutions) was fully characterized, verifying that the antibiofilm activity was the result of a combined repulsion and contact killing mechanisms.

Reviewer 2 Report

In manuscript “Innovative antibiofilm smart surface against Legionella for water systems” Filice et al. investigate the effect on L. pneumophila of polypropylene filters coated with the polymer s-PBC with respect to growth inhibition and the biofilm formation.

The zone inhibition and biofilm formation tests showed an effect by inhibition of the growth of L. pneumophila and inhibited L. pneumophila biofilm formation due to the coating polymer s-PBC. The effect is rather clear and explanations of the effect are described and discussed. The results are interesting and should be further investigated.

It is important to investigate new surfaces that could inhibit or reduce biofilm formation, which is a prerequisite for Legionella growth.

Since L. pneumophila generally does not form biofilm in natural systems – but is dependent on the presence of a biofilm and protozoans, it would have been interesting if the effect on natural biofilm formation were investigated. 

Comments:

1) The limitations of the study should be discussed further, the study is with laboratory-grown L. pneumophila in an artificial laboratory system. There is missing information on the effect of the s-PBC surface coating on natural more complex biofilm formation.

2) It is not clear when and where the smart surface could be used. In the Introduction water filters (point-of-use filters) are mentioned, the present investigation does not point at an improved effect of using the coating on filter material, apart from reducing the pore size – see below. A comparative test with existing filters would improve the study. 

3) 2.4. Filter coating test: What is the pore size of the PP filter? 

4) 2.4. Filter coating test & Results: The procedure is not clear: “Five filtrations of 50 mL and one last filtration of 200 mL of L. pneumophila culture were performed.”

Were it five filtrations of 50 mL with the same filter (respectively for PP and s-PBC@PP), and was an aliquot of 0.1 mL for each filtrate cultured (Figure 5 first a/bII and fifth a/bIII)? Were the filtrates of the 5 x 50 mL filtrations pooled and the sixth filtrations of 200 mL performed with this pool (still same filter) and 0.1 mL of the filtrate cultured, Figure 5 cI and cII

5) Results: As stated in the Result section, the retention – or removal efficiency is probably due to a mesh reduction due to the polymeric coating. Then the efficiency of the coating is probably attributed to reduced pore size and not due to the specific effect of the s-PBC in reducing the survival of L. pneumophila. This effect might be obtained more efficiently by the use of commercially available point-of-use filters with defined pore size; this should be mentioned and discussed. 

6) It would have been interesting with colony counts – comparing the CFU/L in the starting solution, the filtrated water, and in the retention – the sum of the colonies (CFU/L) in the filtrate and retention should be matched with the colony count (CFU/L) in the starting solution. If the result shows a significantly lower count for retention and filtrates for the s-PBC polymer-coated PP than for PP, it could indicate an extra effect (killing) – than just retention – of the s-PBC polymer. 

7) The aim of the filtration study should be mentioned, as it is not clear. The results should be discussed further. Were the results as expected or were an effect besides retention (based on the results for the inhibition test and biofilm formation test) expected?     

Author Response

In manuscript “Innovative antibiofilm smart surface against Legionella for water systems” Filice et al. investigate the effect on L. pneumophila of polypropylene filters coated with the polymer s-PBC with respect to growth inhibition and the biofilm formation.

The zone inhibition and biofilm formation tests showed an effect by inhibition of the growth of L. pneumophila and inhibited L. pneumophila biofilm formation due to the coating polymer s-PBC. The effect is rather clear and explanations of the effect are described and discussed. The results are interesting and should be further investigated.

It is important to investigate new surfaces that could inhibit or reduce biofilm formation, which is a prerequisite for Legionella growth.

Since L. pneumophila generally does not form biofilm in natural systems – but is dependent on the presence of a biofilm and protozoans, it would have been interesting if the effect on natural biofilm formation were investigated. 

Comments:

1) The limitations of the study should be discussed further, the study is with laboratory-grown L. pneumophila in an artificial laboratory system. There is missing information on the effect of the s-PBC surface coating on natural more complex biofilm formation.

Authors thank the referee for this helpful comment. A statement underlying the limitations and the future perspectives of the study has been added in lines 358-360.

2) It is not clear when and where the smart surface could be used. In the Introduction water filters (point-of-use filters) are mentioned, the present investigation does not point at an improved effect of using the coating on filter material, apart from reducing the pore size – see below. A comparative test with existing filters would improve the study. 

Authors apologize if the information was not properly reported. As added in the Introduction section (lines 95-98), the aim was to preliminary assay the antibiofilm activity of polymer s-PBC towards Legionella by inhibition, using in vitro biofilm formation and water filtration tests. Water filtration tests, in particular, were used as representative of possible applications as coating in water systems. In fact, the need for methods to reduce Legionella colonization of pipes requires the develop of single or multi-level functionalization steps over surfaces, preferably before the biofilm is formed [17].

3) 2.4. Filter coating test: What is the pore size of the PP filter? 

The PP filters are composed of fibers with diameter ranging between 0.2 and 5 micron. They form a three-dimensional network where the pore size has a wide range distribution. Then, it is more appropriate to define a porosity instead of a pore size. The filter porosity eF is 0.98 and was determined using the formula: eF= 1- rSF/(rF·L), where rSF is the surface density calculated as the mass of the filter divided by the surface area (mF/AF); rF is is the fibers’ material density (910 Kg/m3 for polypropylene was used); L is the PP filter thickness. Authors added the information in lines 213-219.

4) 2.4. Filter coating test & Results: The procedure is not clear: “Five filtrations of 50 mL and one last filtration of 200 mL of L. pneumophila culture were performed.”

Were it five filtrations of 50 mL with the same filter (respectively for PP and s-PBC@PP), and was an aliquot of 0.1 mL for each filtrate cultured (Figure 5 first a/bII and fifth a/bIII)?

Authors apologize if data were not clearly reported. An aliquot of 0.1 mL for each filtrate (first and fifth with PP and s-PBC@PP filters) has been used for the culture. A sentence clarifying this procedure has been inserted in lines 309-310.

Were the filtrates of the 5 x 50 mL filtrations pooled and the sixth filtrations of 200 mL performed with this pool (still same filter) and 0.1 mL of the filtrate cultured, Figure 5 cI and cII

The filters used for each filtration (first to sixth) were the same. No filtration was pooled and filtered again, including the 200 mL filtration. The sentence in lines 319-320 has been modified as follows:

Before - “The same result was observed for the filtration tests conducted with the same filters using a 200 mL water sample

After- “The same result was observed for the filtration tests conducted with the same filters using another 200 mL aliquot from the starting water sample”.

5A) Results: As stated in the Result section, the retention – or removal efficiency is probably due to a mesh reduction due to the polymeric coating. Then the efficiency of the coating is probably attributed to reduced pore size and not due to the specific effect of the s-PBC in reducing the survival of L. pneumophila.

The aim of the work was to test the antibiofilm activity of the s-PBC. This was verified by testing the anti-attachment process of planktonic Legionella on top of the modified filter surface and that, although randomly deposited by gravity, bacteria were inhibited in their proliferation process. This statement has been added in lines 364-368.

5B) This effect might be obtained more efficiently by the use of commercially available point-of-use filters with defined pore size; this should be mentioned and discussed. 

In the Conclusions section it has been specified that another possible application of the proposed polymer could be the coating of pipes as antibiofilm surface (lines 362-363).

6) It would have been interesting with colony counts – comparing the CFU/L in the starting solution, the filtrated water, and in the retention – the sum of the colonies (CFU/L) in the filtrate and retention should be matched with the colony count (CFU/L) in the starting solution. If the result shows a significantly lower count for retention and filtrates for the s-PBC polymer-coated PP than for PP, it could indicate an extra effect (killing) – than just retention – of the s-PBC polymer. 

Authors thank the referee for this helpful comment. Authors’ idea was not to demonstrate a killing effect of s-PBC towards L. pneumophila but to verify its inhibition properties against the bacterial biofilm formation. Therefore, colony counts of the CFU/L in starting and retention solutions were not performed and matched. This explanation has been added in lines 353-357.

7) The aim of the filtration study should be mentioned, as it is not clear.

Authors clarified the aim of the filtration study in lines 135-136.

The results should be discussed further. Were the results as expected or were an effect besides retention (based on the results for the inhibition test and biofilm formation test) expected?     

Also results from the study have been detailed discussed in lines 358-362.

Reviewer 3 Report

Review of the manuscript microorganisms-1642489entitled “Innovative antibiofilm smart surface against Legionella for water systems.”. In this manuscript authors evaluated the antibiofilm activity of a new smart surface using different experimental approaches. The topic is very interesting and new. Moreover the manuscript is clear and well written.

I have only some minor suggestion:

In the abstract and in the aim of the study Introduction) also the filter coating test should be reported besides the “Biofilm formation test” and the the “Zone of inhibition test”.

How many replicates were performd for each test? This information should be added in the materials and methods section.

Page 7 line 255 – Results or discussion?

Discussion section should be implemented.

Table 1: Associate each line with the specific test

It is not clear the sequential filtration of water samples inoculated. Futhe information should be added in the materials and methods section.

Author Response

Review of the manuscript microorganisms-1642489entitled “Innovative antibiofilm smart surface against Legionella for water systems.”. In this manuscript authors evaluated the antibiofilm activity of a new smart surface using different experimental approaches. The topic is very interesting and new. Moreover the manuscript is clear and well written.

I have only some minor suggestion:

  1. In the abstract and in the aim of the study Introduction) also the filter coating test should be reported besides the “Biofilm formation test” and the the “Zone of inhibition test”.

Authors thank the referee for the positive consideration of the proposed work. Authors’ aim for filtrations study has been included in both the Abstract (lines 36-37) and Introduction (lines 135-136) section as suggested.

  1. How many replicates were performd for each test? This information should be added in the materials and methods section.

Authors agree that this information has not been included. However, the polymer inhibition properties have been already tested in a previous work (see ref [25]), where several replicas have been performed in order to characterize the response of bacteria after s-PBC exposure. The reported data concerned a mechanism that can be potentially inferred to all bacteria species, as proved for Legionella and Pseudomonas.

  1. Page 7 line 255 – Results or discussion?

Authors apologize for the mistake. The word “discussion” has been removed.

  1. Discussion section should be implemented.

Authors included more details, especially concerning the filtration tests and further improvements of the research, in lines 351-369.

  1. Table 1: Associate each line with the specific test

Authors apologized for the confusing information. The pH values reported in Table 1 concerned all media involved in the same biofilm formation experiment, i.e. the Legionella growth in chamber slides, after exposure to floating coupons, and tap and distilled water used in the coupon preparation step of the zone of inhibition test. A specification of the media applications has been included in the manuscript (lines 282-283),

  1. It is not clear the sequential filtration of water samples inoculated. Futhe information should be added in the materials and methods section.

Authors apologize if the procedure was not totally clear. Two sentences adding more details for a better comprehension of the filtration tests have been added in lines 309-310 and 319-320.

Reviewer 4 Report

The authors cope with an issue of great importance, that of biofilm formation. Legionella causes thousands of human cases each year. There are several methods followed trying to reduce its population in the water. The greatest enemy against these methods is the formation of biofilm since it can protect Legionella from the disinfectants used.

The authors propose a NexarTM-modified surface for smart L. pneumophila removal and simultaneous prevention of its biofilm formation in water systems. The methodological part is well defined and the results seem well presented. The proposed technique showed a significant potential in reducing planktonic and sessile (biofilm) Legionella cells and could be used as an inactivation system for avoiding biofilm formation.

Perhaps the authors could explain why they used serogroup 2-15 instead of L. p. 1 which is of greater importance and they could also expand their discussion putting some more data from other published studies.

Author Response

The authors cope with an issue of great importance, that of biofilm formation. Legionella causes thousands of human cases each year. There are several methods followed trying to reduce its population in the water. The greatest enemy against these methods is the formation of biofilm since it can protect Legionella from the disinfectants used.

The authors propose a NexarTM-modified surface for smart L. pneumophila removal and simultaneous prevention of its biofilm formation in water systems. The methodological part is well defined and the results seem well presented. The proposed technique showed a significant potential in reducing planktonic and sessile (biofilm) Legionella cells and could be used as an inactivation system for avoiding biofilm formation.

Perhaps the authors could explain why they used serogroup 2-15 instead of L. p. 1 which is of greater importance and they could also expand their discussion putting some more data from other published studies.

Authors thank the referee for the good evaluation of the proposed work. Authors’ aim was to test the antibiofilm properties of the proposed polymer starting from environmental strains because, as added at the end of the Introduction section, lines 125-134, “L. pneumophila SG 1 to 10, 12, 13, and 15 have always been isolated from water distribution systems worldwide more frequently than the other serogroups [29–31]. Although environmental strains (SG 2 to 15) account for only 16 to 20% of legionellosis cases [32], there is evidence that patients with L. pneumophila of SG 2 to 15 show typical symptoms of Legionella pneumonia, but Legionella urinary antigen tests (UATs), which are rapid tools for early diagnosis of legionellosis, are negative because UATs detect only SG 1 [33–35]. In a Public Health perspective, this is noteworthy be-cause there is an association among delayed laboratory diagnosis, therapy and prognosis. In particular, the delay in appropriate therapy of legionellosis is associated with increased mortality [36]. For these reasons, we decided to test only the environmental strains (SG 2 to 16).

Round 2

Reviewer 1 Report

Thank you for  responding to comments. A small note, the names of the species Legionella should be in italics (for the text in red).